# Sulfate Resistance in Cements Bearing Ornamental Granite Industry Sludge

**DOI:** 10.3390/ma13184081

**Published:** 2020-09-14

**Authors:** Gabriel Medina, Isabel F. Sáez del Bosque, Moisés Frías, María Isabel Sánchez de Rojas, César Medina

**Affiliations:** 1Departamento Ingenierías Mecánica, Informática y Aeroespacial, Escuela de Ingenierías Industrial e Informática, Campus de Vegazana, Universidad de León, s/n, 24071 León, Spain; gmedinaia@gmail.com; 2Departamento de Construcción, Escuela Politécnica de Cáceres—Grado de Ingeniería Civil, Universidad de Extremadura, Instituto de Investigación de Desarrollo Territorial Sostenible (INTERRA), 10003 Cáceres, Spain; isaezdelu@unex.es; 3Departamento de Cementos y Reciclado de Materiales, Instituto de Ciencias de la Construcción Eduardo Torroja (IETcc-CSIC), Sostenibilidad en Materiales de Construcción, Universidad de Extremadura, Unidad Asociada al CSIC, 28033 Madrid, Spain; srojas@ietcc.csic.es; 4Departamento de Construcción, Escuela Politécnica de Cáceres—Grado de Ingeniería Civil, Universidad de Extremadura, Sostenibilidad en Materiales de Construcción, Universidad de Extremadura, Unidad Asociada al CSIC, Instituto de Investigación de Desarrollo Territorial Sostenible (INTERRA), 10003 Cáceres, Spain

**Keywords:** sulfate, granite dust, durability, binary cements, performance

## Abstract

This study explores the effect on sulfate resistance of the use of ornamental granite industry waste as a supplementary cementitious material (at replacement ratios of 10% and 20%) in cement manufacture. The present paucity of scientific knowledge of the behaviour of these new cements when exposed to an external source of sulfates justifies the need for, and the originality of, this research. After characterising the waste chemically and mineralogically, cement paste specimens were prepared in order to determine the durability of the newly designed eco-cements using Köch–Steinegger corrosion indices. The new hydration products, which might induce microstructural, mineralogical, or morphological decay in the specimens, were also analysed by comparing the samples before and after soaking in a sodium sulfate solution for different test periods. Respect to the results, the damage to pastes bearing 10% granite sludge (GS) is the same as observed in OPC, whilst the former exhibit a higher Köch-Steinegger corrosion rate (1.61) than both OPC and OPC+20GS. Soaking the pastes in sodium sulfate induces matrix densification due to ettringite formation and gypsum precipitation in the pores. Further to those results, at an optimal replacement ratio of 10%, these alternative, eco-friendlier materials can be used in the design and construction of non-structural cement-based (mortar or concrete) members exposed to an external source of sulfate.

## 1. Introduction

The ornamental stone industry generates large volumes of waste (~58% of total output). Quarrying yields solid refuse (20% to 22% of total output) and solid particles that result from cutting and polishing, mixed with the water needed to cool the cutters, form sludge (20% to 30% of total output) [1] subsequently drained and deposited in sludge ponds. When the ponds fill up, they are emptied and the waste is removed to uncontrolled muck pits or spoil banks (Figure 1), with the concomitant adverse environmental impacts. Such sludge reduces the soil permeability to the detriment of plant growth, whilst the particulate matter (comprising primarily silica) released into the air as the water evaporates under atmospheric action (temperature and wind) may cause severe harm to human beings in the form of diseases such as silicosis [2,3]. 

The valorisation of that waste, a significant challenge to modern society, is fully aligned with the European Commission’s recently adopted ‘New circular economy action plan for a cleaner and more competitive Europe’ [4]. That plan is designed to hasten the transition to a regenerative growth model that returns to the planet more than it takes from it, maintain resource consumption within planetary limits, and double the rate of circular material used: in short, to further sustainability and the use of secondary raw materials. Earlier authors characterising this type of waste based on its chemical (oxide) composition found that it comprises primarily SiO_2_ (45% to 70%) [3,5,6,7,8,9] and Al_2_O_3_ (10% to 18%) [3,5,6,7,9,10], small amounts (usually <10%) of CaO [3,5,6,8,9], and variable amounts of Fe_2_O_3_, generally ≤10% [3,6,8,9], although values of ≥20% have also been reported [5]. The common denominator in most such granite waste is that the sum of SiO_2_+Al_2_O_3_+Fe_2_O_3_ accounts for 70% or over of the total [3,5,6,8,9]. 

Granite sludge mineralogical composition, in turn, is made up essentially of quartz [5,6,7,8,9,11]; along with potassium feldspars [5,6,11], such as microcline [7,8] and orthoclase [9]; plagioclase or sodium feldspars [5,6,7,11], mainly albite [8,9]; micas, mostly muscovite [9,11] and biotite [5,9,11]; and, clays, including kaolinite [6,8,9] and illite [8]. Calcite [5], hematite [7,9], magnetite [8], and chlorite minerals [9] have also been detected. Using a saturated lime solution to measure granite sludge pozzolanicity, Medina et al. [9] found the initially low values to be unaffected by heating the material to 600 °C to 700 °C.

Some researchers have addressed ornamental granite sludge from the perspective of reuse as a raw material in new cement-based or masonry materials [6,8,12,13,14]. Of the former, some studied application of this waste as coarse or fine aggregate in concrete [3,7,15,16,17,18,19] or as a partial cement replacement in concrete [20,21] or mortar [1,22] manufacture. 

In a review paper, Rana et al. [23] noted that, where recycled granite aggregate was used to fully replace conventional aggregate, the resulting product exhibited similar or even greater strength than conventional concrete. Where granite dust was used to replace fines or as a filler, the compressive strength was also generally comparable to or greater than in conventional concrete at replacement ratios of up to 50% [24,25], although the range that was established by some authors for the optimal percentage to ensure higher than conventional material strength was 15% [7] to 20% [3]. Yet, others defined a much higher 70% as the optimal value. Where granite sludge is used as a partial cement substitute, compressive strength in the recycled products has been observed to be similar to or slightly greater than in the conventional materials at replacement ratios of 5% [20] in concrete, 5% to 10% in masonry mortar [5], and 15% in standardised mortars [26]. Lower strength has been reported for granite waste ratios of 20% or higher in masonry [5], standardised [9,26], mortars, and pastes [27].

Nonetheless, at this time little is known regarding the durability of new construction materials bearing granite sludge as a cement replacement. An understanding of their water sorptivity, capillarity, and behaviour in aggressive chloridic, carbonatic, or sulfatic environments are areas meriting attention from the scientific community, for they have a direct effect on material performance during its service life and the structures present in its composition. The durability studies conducted to date, focusing on carbonation [28], shrinkage [29], and water transport [28,29,30], have reported that: (i) for ratios of up to 15%, carbonation depth was up to 21.4% shallower than in new conventional mortars [28]; (ii) shrinkage declined at the rate ≤20% [28,29]; and, (iii) total water absorption was ~15% greater in mortars bearing 20% sludge than in conventional mortars [29,30]. Nothing has yet been published internationally on the sulfate resistance of cements bearing granite sludge as a replacement, however. The only two studies conducted on sulfate resistance in the presence of granite sludge referred to use of the waste in concrete: in one, as coarse aggregate [18] and in the other as fines [7]. The former reported higher and the latter lower sulfate resistance. In that same vein, the scant research focusing on the durability of mortars containing (<25 wt%) ornamental marble [23,31] or (<15 wt%) quartzite sludge [32] found that sulfate resistance was enhanced with the use of marble, whereas no decline in expansion was observed for quartzite. 

Against that backdrop, this study provides scientific-technical insight into the effect of replacing 10% or 20% cement with granite sludge on sulfate resistance by analysing the mechanical behaviour, porosity, and weight of cement pastes made with new blended cements exposed to aggressive environments for different times. The microstructural changes taking place due to the chemical attack prompted by soaking in sodium sulfate were determined while using four instrumental techniques: mercury intrusion porosimetry (MIP), X-ray diffraction (XRD), Fourier transform infrared spectroscopy (FTIR), and scattering electron microscope (SEM/EDX). 

## 2. Materials and Methods

### 2.1. Materials

The granite waste used, furnished by an ornamental granite plant at Quintana de la Serena in the Spanish province of Badajoz, consisted dust generated during cutting, which, mixed with the water needed to cool the cutters, forms sludge. When oven-dried in laboratory at 100 °C, the granite sludge (GS) yielded a fine powder with a particle size of under 90 µm. Analyses conducted in an earlier study [9] showed the chemical composition of the granite waste, analysed with XRF, to comprise essentially SiO_2_+Al_2_O_3_ (~85 wt%) with a low (2.36 wt%) CaO content, whilst its XRD identified quartz, feldspars, phyllosilicates, and hematite in its mineralogy. The reactive silica content was 22.4% [9], a value slightly lower than the 25% defined for natural pozzolans in European standard EN 197-1.

The EN 197-1-compliant CEM I 42.5 R portland cement (OPC) used was supplied by a Lafarge Group plant at Villaluenga de la Sagra in the Spanish province of Toledo.

### 2.2. Blends

The new cements, blended in a high-speed power mixer to ensure uniformity, comprised OPC+GS, with GS comprising 10% or 20 wt% of the total. Those values lay within the 6% to 20% range for cement type II/A and 11% to 35% for cement type IV/A of the aforementioned standard EN 197-1 [10]. The physical, mechanical, and chemical properties of the new cements that are given in Table 1 show that, irrespective of the replacement ratio, they met all of the requirements laid down in EN 197-1 [10] for ordinary cements.

### 2.3. Methodology

The flow chart presented in Figure 2 describes the five steps comprising this study: step 1, laboratory pre-conditioning (oven-drying) of the waste; step 2, waste characterisation: step 3, blending of component materials; step 4, cement paste mixing and curing; and, step 5, paste exposure to the aggressive medium and subsequent characterisation.

In step 4, the OPC, OPC+10SG, and OPC+20SG pastes were mixed with deionised water at a water/cement ratio of 0.5 to prepare 1 × 1 × 6 cm^3^ prismatic specimens, 12 each per mix, medium (sulfates and water), and exposure time. They were demoulded after 24 h and subsequently cured for 21 d at 100% relative humidity and a temperature of 20 ± 1 °C. In step 5, groups of 12 specimens were then soaked in an aggressive 0.3 M sodium sulfate solution (4,4 wt% Na_2_SO_4_ at a liquid/solid volume ratio of 22) or deionised water as the reference for 14 days, 56 days, 90 days or 180 days at 20 °C, based on the Köch–Steinegger method [33]. 

At each test age, the specimens were washed three times in deionised water prior to characterisation in order to eliminate any excess salts and dried to a constant weight in a laboratory oven at 40 °C.

Likewise, in step 5, specimen flexural strength and variation in weight were determined at each exposure time and pore size distribution was analysed in the 56 days and 180 days exposed specimens. The microstructural analysis tests were supplemented with XRD, FTIR, and SEM/EDX identification of the new components formed.

### 2.4. Instrumental Techniques

Sample mineralogy was determined on a Bruker AXS D8 X-ray powder diffractometer (Bruker Corporation, Madrid, Spain) fitted with a 3 kW (Cu Kα) copper anode and a wolfram cathode X-ray generator. The scans were recorded between 2θ angles of 5° to 60° at a rate of 2°/min. The voltage generator tube operated at a standard 40 kV and 30 mA [34].

The materials were characterised on a Thermo Scientific Nicolet 600 Fourier transform infrared spectrometer (Thermo Scientific Corporation, Madrid, Spain) featuring a spectral resolution of 4 cm^−1^ across a range of 4000–500 cm^−1^ [34].

The Hitachi S4800 electron microscope that was used to study the morphology of the 180 days blended cements exposed to the aggressive medium was coupled to a Bruker Nano XFlash 5030 silicon drift detector (Micromeritics Instrument Corp., Aachen, Germany) for EDX determination of the chemical composition of the samples [34].

Porosity was quantified on a Micromeritics Autopore IV 9500 mercury porosimeter (Norcross, GA, United States) designed to measure pore diameters of 0.006 µm to 175 μm and operate at pressures of up to 33,000 psi (227.5 MPa) [35].

Mechanical strength was found on an Ibertest Autotest 200/10-SW test frame that was fitted with an adapter for 1 × 1 × 6 cm^3^ specimens [34].

## 3. Results & Discussion

### 3.1. Specimen Variation in Mass with Soaking Time

Weight was observed to rise in all specimens, irrespective of whether they were soaked in water or sulfates (Figure 3). Weight gain was calculated, as shown in Equation (1), where: Δ*W* is variation in weight; *m_0_* is weight prior to exposure (21 d-cured specimens); and, *m_i_* is weight at exposure time ‘i’ (t_i_ = 14 days, 56 days, 90 days, or 180 days).
(1)$$∆W=\left[\left(m_{i}-m_{0}\right)/m_{0}\right]\times 100$$

The weight gain in the water-soaked specimens was directly associated with anhydrous cement particle hydration and the concomitant generation of hydration products that densified the cementitious matrix. The gain was greater in the OPC + 10GS and OPC + 20GS specimens due to the effect of a simultaneous second reaction: GS particles interacted with portlandite (with which they exhibit slow pozzolanicity [27]), yielding C-S-H gels and further densifying the matrix. The weight gains associated with sulfate attack were defined as the difference between the gains in the sulfate- and water-soaked specimens (Figure 3). Those gains, which were observed to be similar in the three types of binder (0.22% for OPC, 0.23% for OPC + 10GS and 0.25% for OPC + 20GS), were attributable to the reaction between hydrated cement phases and sulfate ions penetrating the pore system from the medium. As both gypsum, the primary and ettringite the secondary product of that reaction [36] occupied greater volume than the cement phases and ions, pore volume declined, raising matrix density. 

Those findings were consistent with the increase in weight that was recorded by other researchers analysing resistance to a Na_2_SO_4_ solution by conventional type I cements/mortars; type I cements/mortars bearing (<30 wt%) fly ash [37,38], (30 wt%) construction and demolition waste (CDW)-sourced masonry materials [34], or (20 wt%) fired-clay sanitary ware industry polishing and enamelling waste [39]. Such weight gains attest to the high cement deterioration potential of Na_2_SO_4_.

### 3.2. Cement Paste Surface 

A macroscopic inspection (Figure 4) of the 1 × 1 × 6 cm^3^ specimens that were exposed to the Na_2_SO_4_ solution showed no outer deterioration in the OPC or OPC + 10GS samples, whereas the OPC + 20GS were cracked and scaling, particularly at and around the edges. The former two pastes would qualify as category 0 (no damage) and the latter as category Mi (minor damage) in the Mittermayr et al. [40] classification of mortars exposed to sulfate attack. 

The damage observed was directly associated with the expansive nature of the products of cementitious matrix—sulfate interaction, a process in which initial cracking (internal stress > matrix tensile strength), followed by swelling, scaling, and detachment [41] compromises cement system durability.

### 3.3. Sulfate Resistance

Sulfate resistance was determined with the expression for corrosion resistance that was proposed by Köch–Steinegger (Equation (2)):(2)$$\mathrm{CI}=F_{\mathrm{SS}}/F_{\mathrm{SW}}$$
where CI is corrosion index; Fss flexural strength at aggressive sulfate exposure time ‘i’; and, Fsw sulfate resistance in water-soaked specimens at the same exposure time. 

Strength was higher in the latter than the former at all exposure times, according to the flexural strength data for the water (F_SW_) and sulfate (F_SS_)-soaked pastes listed in Table 2. That circumstance was directly related to the inverse linear relationship (Figure 5) between flexural strength (FS) and pore system properties (total porosity and percentage of macropores), i.e., lower total porosity and macropore volume translated into higher FS. The use of granite sludge (GS) induced a slight decline in flexural strength due to the smaller C-S-H content in cementitious pastes OPC+10GS and OPC+20GS relative to the reference, and greater porosity in the experimental materials, as discussed in Section 3.4 [27].

The bar graph presented in Figure 6 shows that the corrosion index for the pastes declined linearly with rising Na_2_SO_4_ exposure time (correlation coefficient, R^2^, >0.809, denoting the effect of sulfate chemical action on cement-based material performance. In the Köch–Steinegger method, pastes are deemed sulfate-resistant when their 56 days corrosion index is greater than or equal to 0.70. Further to that criterion, the cementitious matrices bearing the new cements studied here behaved satisfactorily when exposed to Na_2_SO_4_, with OPC + 10GS exhibiting a CI value of 1.61 and OPC + 20GS of 1.58. Nonetheless, at all exposure times, except 14 days, the OPC + 20GS pastes had a lower CI than the OPC and OPC + 10GS materials, with the difference ranging from 1% to 11%. OPC + 10GS behaved comparably to OPC, with minor rises in CI of 1.5% to 1.8%. In light of those findings, 10% GS would be the optimal replacement ratio to ensure durability.

The corrosion index observed for OPC lay within the 1.29 to 2.50 range that was observed in earlier studies. The values for OPC + 10GS and OPC + 20GS were: (i) similar to those found for binary cementitious matrices bearing either 15% silico-manganese slag (CI = 1.49) [42] or 20% fired clay product polishing and enamelling waste (CI = 1.48) [39]; or, ternary matrices with 21% paper sludge + fly ash (CI = 1.55) [43]; and, (ii) lower than in pastes containing construction and demolition waste masonry materials (CI = 2.30) [34].

### 3.4. Variation in Pore Structure

Table 3 lists the pore system properties in 56 days and 180 days sulfate-soaked pastes and Figure 7 plots the effect of soaking on pore size. 

As the data in Table 3 show, total porosity declined with exposure time by 18.7% in OPC, 20.9% in OPC + 10GS, and 24.7% in OPC + 20GS. The steeper decline in the samples bearing granite sludge was associated with their late-age pozzolanicity [27] and the precipitation of expansive compounds gypsum and ettringite in the pore system [44] that reduced permeability and retarded deterioration [45]. 

The variation in total porosity recorded here for 180 days soaking in Na_2_SO_4_ was slightly wider than that observed by Goñi et al. [43] between type I cement (13.9% to 8.8%) and ternary binders bearing 21 wt% paper sludge + fly ash (17.5% to 11.5%). In contrast, the observed 56 days total porosity was similar to the values that were reported by Frías et al. (~22 to 25%) [42] for cementitious systems with 5% or 15% silico-manganese slag able to fix lime at a rate similar to the rate observed for GS [9]. 

Table 3 also shows that the new pastes (OPC + 10GS and OPC + 20GS) were more porous than OPC, irrespective of exposure time, as previously observed by Liu et al. [45] in cements bearing 20% or 40% fly ash.

The pore size distribution curves presented in Figure 6 show that sulfate exposure refined the pore system, inducing a decline in 180 days mean diameter of 25.0% in OPC, 34.3% in OPC + 10GS, and 28.0% in OPC + 20GS. As noted by other researchers [43,44], decline is associated with a reduction in macropore volume and rises in the capillary (0.01 µm to 0.05 µm) and small capillary (0.002 µm< Φ < 0.01 μm) fractions. Such refinement was also consistent with the findings reported by Asensio et al. [34], who also observed a shift in the curve to smaller diameters due to the precipitation of new compounds in the pore system and CDW masonry material pozzolanicity. 

### 3.5. Composition and Microstructure 

The XRD patterns for water- and 0.3 M Na_2_SO_4_-soaked pastes of OPC, OPC + 10GS, and OPC + 20GS showed that the samples that were soaked in sulfate for 180 days had more intense reflections for ettringite and gypsum (Figure 8). Those two primary deterioration products that were associated with sulfate attack were also observed by Liu et al. [46] in portland cement pates and Veiga and Gastaldini [47] in cements bearing granulated blast furnace slag. Calcite was also identified in the form of carbonate resulting from carbon dioxide dissolution in the medium [48], along with crystalline compounds, such as anhydrous cement (belite) and the orthoclase, quartz, and muscovite present in unreacted GS [9].

The FTIR spectra for the 180 days water- and sulfate-soaked pastes reproduced in Figure 9 corroborate the XRD findings. The band at 3641 cm^−1^ in these spectra, essentially associated with portlandite, a hydration product, was less intense in the pastes soaked in the sulfate solution for it reacted with the sulfates to yield gypsum. The band at 1118 cm^−1^, attributed to the ν_3_ vibrations generated by the SO_4_^2−^ in gypsum and ettringite, was also more intense on the sulfate-soaked paste diffractograms [49]. The weak band at 1045 cm^−1^ is characteristic of gypsum ν_3_ vibrations and both the weak band at 608 cm^−1^ and the medium intensity signal at 668 cm^−1^ of its ν_4_ vibrations. No variation was detected in the band associated with calcium silicate hydrate (C-S-H) at 979 cm^−1^ to 980 cm^−1^ [50]. The bands that were associated with ν_3_ vibrations generated by the CO_3_^2−^ group in carbonates, at 1485 cm^−1^ and 1431 cm^−1^, were also observed to remain unchanged [49,51].

After the pastes were soaked for 180 days in Na_2_SO_4_, one of the most prominent hydration products formed was ettringite (AFt) (Figure 10). Its long needled, ‘hedgehog-like’ structures [52,53] were primarily observed inside pores (Figure 10a–d) and in cracks where AFt normally precipitates [54] as a result of the pressure conditions prevailing and ions present in those areas, especially the Al(OH)^4−^ needed for ettringite formation [55,56].

The presence of gypsum plates, a product of the reaction between SO^2−^_4_ and Ca^2^ ions, is visible in the pore solution in Figure 10e,f.

## 4. Conclusions

The following conclusions can be drawn from this study.

-All of the cement pastes exposed to external sulfates deteriorated further to the mechanism associated with sulfate attack.-The OPC and OPC + 10GS pastes exhibited the same level of damage, whereas deterioration was more severe in paste OPC + 20GS. Therefore, the macroscopic classification for the former two was 0, ‘no damage’, and for the latter Mi, ‘minor damage’.-Microcracking was observed in the 180 days-soaked pastes prepared with 20% GS (OPC + 20GS).-Sulfate and sodium ingress into the paste microstructure translated primarily into ettringite formation inside pores and gypsum plate precipitation, densifying the cementitious matrix.-Matrix densification, which refined the pore system, inducing late age weight gain, was more intense in the pastes bearing GS as a result of the slow pozzolanicity of that material. -Under the present experimental conditions, the Köch–Steinegger corrosion index found for the 56 days-soaked samples was higher than the minimum needed to qualify as sulfate resistant. -Strength under sulfate attack (corrosion index) was higher in the OPC + 10GS pastes than in OPC, where it was higher than in paste OPC + 20GS.-OPC + 10GS cements can be used as alternative binders in the design and construction of non-structural cement-based (mortar or concrete) members that were exposed to an external source of sulfates. 

The findings showed that this waste can be viably used in the cement industry without compromising the durability of the base materials bearing it. That practice would minimise the consumption of natural resources and fuel and lower the associated greenhouse gas emissions, while delivering the benefits inherent in valorising the waste itself. Such eco-favourable results constitute a firm endorsement for the use of granite sludge as an alternative material in the design of sustainable cements.

## 5. Limitations of the Study

New cement sulfate resistance was assessed on the grounds of an accelerated method and sulfate attack associated with sodium cations. Future research would be needed in order to corroborate the findings with other test methods to assess the damage to mortars and concretes as well as material resistance to sources of sulfate that are associated with calcium and magnesium cations. 

## Figures and Tables

**Figure 1 materials-13-04081-f001:**
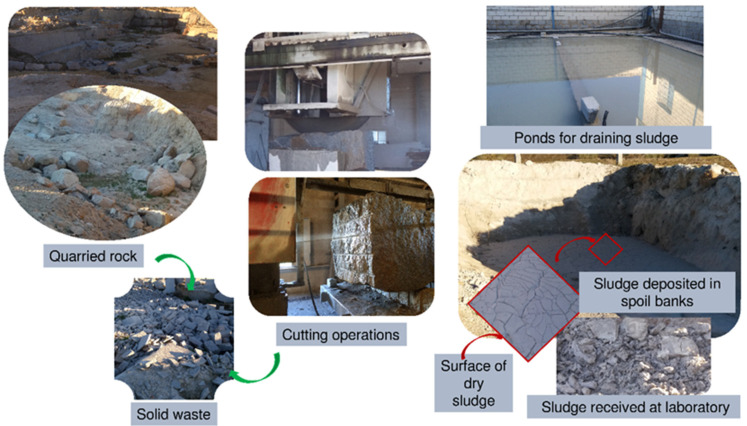
Ornamental granite production and waste generated.

**Figure 2 materials-13-04081-f002:**
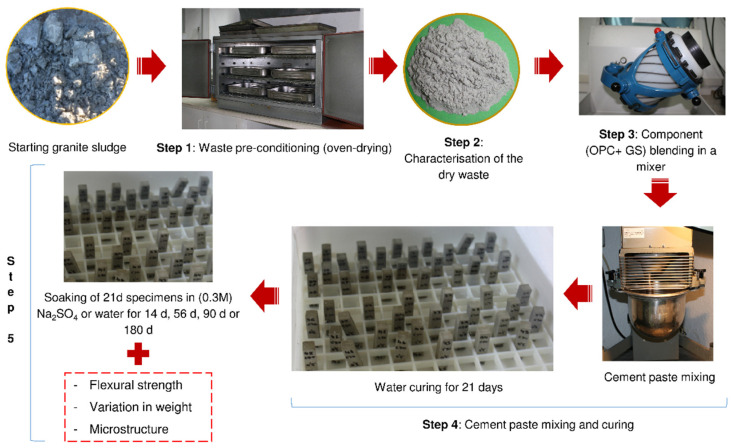
Steps comprising the research conducted.

**Figure 3 materials-13-04081-f003:**
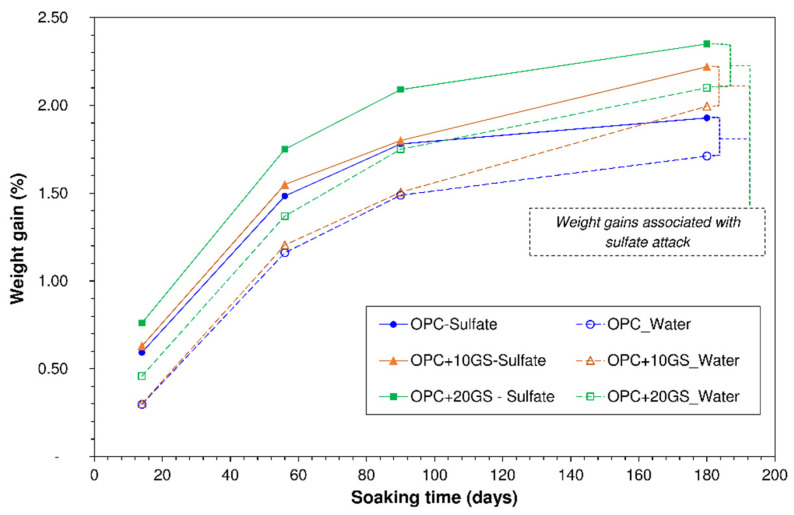
Cement paste weight gain vs. soaking time.

**Figure 4 materials-13-04081-f004:**
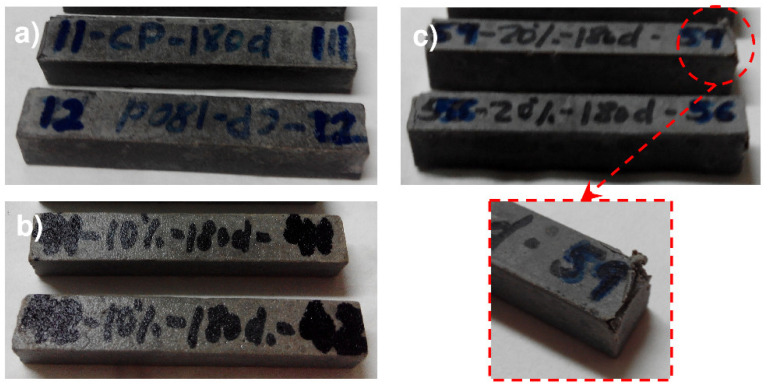
180 days Na_2_SO_4_-soaked specimens: (**a**) OPC; (**b**) OPC + 10GS; and, (**c**) OPC + 20GS.

**Figure 5 materials-13-04081-f005:**
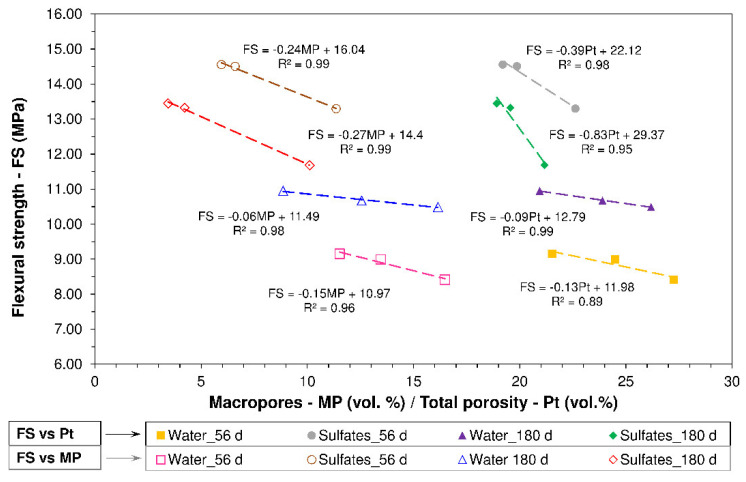
Flexural strength vs. pore system properties.

**Figure 6 materials-13-04081-f006:**
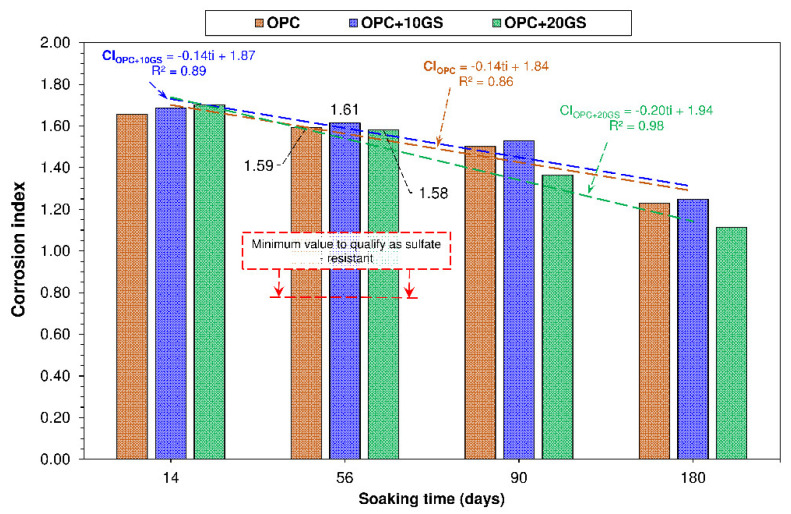
Corrosion index versus soaking time (t_i_).

**Figure 7 materials-13-04081-f007:**
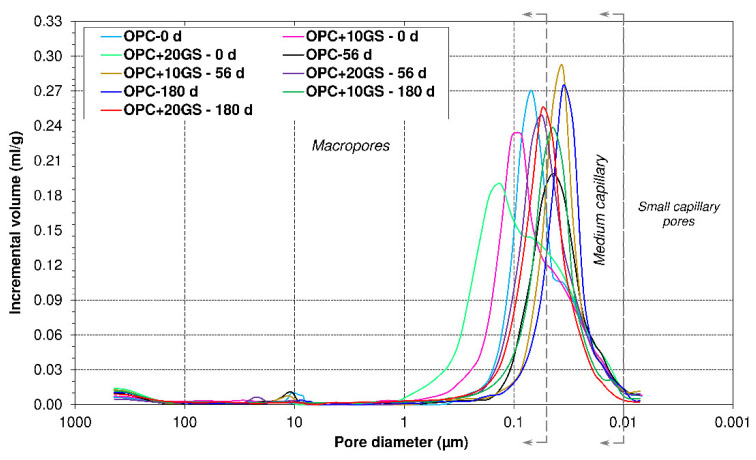
Variation in pore size diameter with time in pastes exposed to sulfates (t_i_ = 0, 56 or 180 days).

**Figure 8 materials-13-04081-f008:**
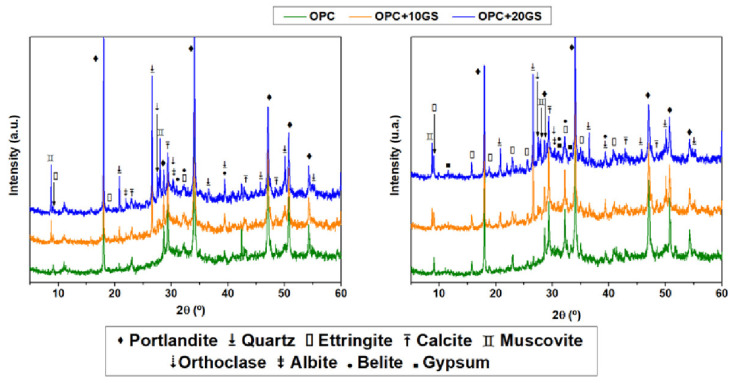
XRD patterns for pastes OPC, OPC+10GS, and OPC+20GS soaked for 180 days in (**left**) water and (**right**) sulfates.

**Figure 9 materials-13-04081-f009:**
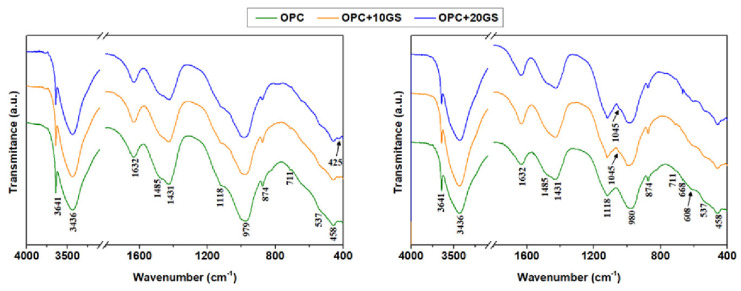
FTIR spectra for pastes OPC, OPC + 10GS, and OPC + 20GS soaked for 180 days in (**left**) water and (**right**) sulfates.

**Figure 10 materials-13-04081-f010:**
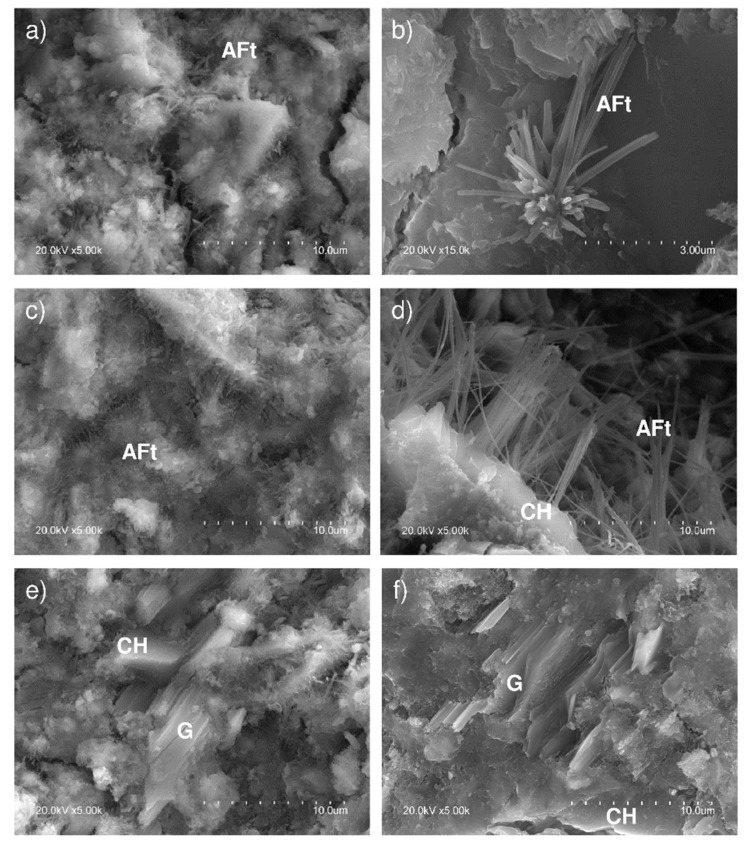
SEM micrographs of cement pastes soaked for 180 days in Na_2_SO_4_: (**a**,**c**) pores filled with ettringite (AFt) needles; (**b**,**d**) detail of AFt needles growing inside pores at the expense of portlandite plate (CH) dissolution; and, (**e**,**f**) gypsum (G) precipitation on portlandite plates.

**Table 1 materials-13-04081-t001:** New cement physical, mechanical, and chemical properties.

Property			Blended Cement			EN 197-1 Requirement
			OPC	OPC + 10GS	OPC + 20GS	
Physical	BET specific surface (m^2^/g)		1.37	1.34	1.32	-
	Initial setting time (min)		210	150	170	≥60.00
	Expansion (mm)		1	1	1	≤10.00
Mechanical	Compressive strength (MPa)	2 day	42.25	37.06	31.19	≥20.00
		28 day	65.67	58.08	51.12	≥42.50
Chemical	Sulfate oxide content (% wt.)		3.14	2.83	2.51	≤4.00
	Chloride content (ppm)		0.01	0.01	0.02	≤0.10
	Pozzolanicity		-	-	Positive	Positive *

***** Standard requirement for type IV cements.

**Table 2 materials-13-04081-t002:** Flexural strength (MPa) in cement pastes exposed to water and sulfates.

Time	Medium	OPC	OPC + 10GS	OPC + 20GS
14	Water	8.87 ± 0.54	8.56 ± 0.35	7.87 ± 0.35
	Sulfates	14.70 ± 0.69	14.44 ± 0.96	13.40 ± 0.88
56	Water	9.15 ± 0.78	8.99 ± 0.75	8.41 ± 0.40
	Sulfates	14.56 ± 0.93	14.51 ± 0.93	13.30 ± 0.89
90	Water	9.56 ± 0.50	9.60 ± 0.66	9.54 ± 0.49
	Sulfates	14.36 ± 0.75	14.68 ± 0.79	13.01 ± 0.96
180	Water	10.95 ± 0.52	10.67 ± 0.64	10.49 ± 0.58
	Sulfates	13.45 ± 0.68	13.32 ± 0.77	11.68 ± 0.75

**Table 3 materials-13-04081-t003:** Pore system properties.

Exposure Time (Days)	Medium	Property	OPC	OPC+ 10GS	OPC + 20GS
0	Water	Pt (vol.%)	23.26	24.74	28.13
		MP (vol.%)	13.59	16.12	19.08
		CP (vol.%)	8.37	8.07	8.40
		MPD (μm)	0.042	0.048	0.054
56	Water	Pt (vol.%)	21.52	24.49	27.26
		MP (vol.%)	11.53	13.46	16.48
		CP (vol.%)	9.62	10.61	10.09
		MPD (μm)	0.040	0.038	0.042
	Sulfates	Pt (vol.%)	19.20	19.87	22.62
		MP (vol.%)	5.95	6.60	11.35
		CP (vol.%)	12.74	12.87	11.02
		MPD (μm)	0.034	0.033	0.040
180	Water	Pt (vol.%)	20.94	23.90	26.19
		MP (vol.%)	8.86	12.56	16.15
		CP (vol.%)	11.74	11.10	9.73
		MPD (μm)	0.034	0.036	0.043
	Sulfates	Pt (vol.%)	18.92	19.56	21.17
		MP (vol.%)	3.45	4.22	10.11
		CP (vol.%)	14.94	15.00	10.99
		MPD (μm)	0.032	0.032	0.039

**Pt**: Total porosity; **MP**: Macropores (Φ > 0.05 μm); **CP**: Capillaries (0.002 < Φ ≤ 0.05 μm); and, **MPD**: mean pore diameter.

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
