# Peer review of "Sulfate Resistance in Cements Bearing Ornamental Granite Industry Sludge"

_materials, 2020, doi:10.3390/ma13184081_

Round 1

Reviewer 1 Report

This paper presents a study on the effect of Ornamental granite industry sludge on sulfate resistance of cement pastes. The topic is interesting and the research paper is well planned and presented. However, there are some comments which can improve the quality of the paper as follow:

  1. Is the replacement was by volume or mass?
  2. Does the size of the granite sludge has any effect on sulfate resistance?
  3. Figures 4-8 are with low resolution and need to be more clear.
  4. English editing is required.

Author Response

The authors wish to thank the reviewer for his/her comments and suggestions, all of which are addressed in the revised MS. All deletions, additions and changes are shown in red in the new version for readier re-review. The paper has also been thoroughly proofread and a number of typos corrected.

This paper presents a study on the effect of Ornamental granite industry sludge on sulfate resistance of cement pastes. The topic is interesting and the research paper is well planned and presented. However, there are some comments which can improve the quality of the paper as follow:

  1. Is the replacement was by volume or mass? The replacement was by mass.
  2. Does the size of the granite sludge has any effect on sulfate resistance? The size of the granite has not negative effect on sulfate resistance, because the specific values of the new blended cement (see table 1) are similar to the cement - OPC (1.37 m2/g).
  3. Figures 4-8 are with low resolution and need to be more clear. The figures 4-8 have a resolution equal to 600 ppp. Nevertheless, we have increased the resolution.
  4. English editing is required. The comment has been considered.

Reviewer 2 Report

The paper entitled “Sulfate resistance in cements bearing ornamental granite industry sludge” explores the effect on sulfate resistance of the use of ornamental granite industry waste as a supplementary cementitious material (at replacement ratios of 10 % and 20 %) in cement manufacture. The findings show that this waste can be viably used in the cement industry, without compromising the durability of the base materials bearing it. That practice would minimise the consumption of natural resources and fuel and lower the associated greenhouse gas emissions while delivering the benefits inherent in valorising the waste itself.

The Ref. [33] (G. Medina, I.F. Sáez del Bosque, M. Frías, M.I. Sánchez de Rojas, C. Medina, “Effect of Granite Waste on Binary Cement Hydration and Paste Performance: Statistical Analysis”, ACI Mater. J., 2019, 116, 63-72), should be added in the Introduction, and the innovation of the submitted paper should be made clearer.

At the end of the “4. Conclusions”, the overall findings could be added (like at the end of the Abstract).

Author Response

The authors wish to thank the reviewer for his/her comments and suggestions, all of which are addressed in the revised MS. All deletions, additions and changes are shown in blue in the new version for readier re-review. The paper has also been thoroughly proofread and a number of typos corrected.

The paper entitled “Sulfate resistance in cements bearing ornamental granite industry sludge” explores the effect on sulfate resistance of the use of ornamental granite industry waste as a supplementary cementitious material (at replacement ratios of 10 % and 20 %) in cement manufacture. The findings show that this waste can be viably used in the cement industry, without compromising the durability of the base materials bearing it. That practice would minimise the consumption of natural resources and fuel and lower the associated greenhouse gas emissions while delivering the benefits inherent in valorising the waste itself.

The Ref. [33] (G. Medina, I.F. Sáez del Bosque, M. Frías, M.I. Sánchez de Rojas, C. Medina, “Effect of Granite Waste on Binary Cement Hydration and Paste Performance: Statistical Analysis”, ACI Mater. J., 2019, 116, 63-72), should be added in the Introduction, and the innovation of the submitted paper should be made clearer. The comment has been considered.

At the end of the “4. Conclusions”, the overall findings could be added (like at the end of the Abstract). The comment has been considered.

Reviewer 3 Report

This manuscript is about Sulfate Resistance in Cements Bearing Ornamental Granite Industry Sludge.

The paper is of average quality. Some suggestions, the authors should consider improving the quality of manuscript:

Abstract:

Add something about the results of the research. [Please add quantities]

 Introduction:

Please add a list of those materials which have already been used in previous studies for same purpose and now Ornamental Granite Industry Sludge will replace. [Please add in line 86-98]

Against that backdrop, this study provides scientific-technical insight into the effect of replacing 10 % or 20 % cement with granite sludge on sulfate resistance by analysing the mechanical behaviour, porosity and weight of cement pastes made with new blended cements exposed to aggressive environments for different times, along with the microstructural changes taking place (using MIP, XRD, FTIR and SEM/EDX). [Please clarify lines 99-103-Clearly explain what and why you are doing this research]

Experimental

Change heading wth standard formate- ‘Materials and Methods’

A comprehensive research framework missing- to follow the research is steps are missing. Add framework-flowchart and write this section in stepwise pattern.

Step.1, Step.2 …..[Please add]

2.1. Materials

Please add details of at least 5-6 materials which were used in previous studies and how authors got an idea to use Ornamental Granite Industry Sludge. Further also explain how it was proved that this material has certain binding properties to replace existing materials upto 10% or 20%. [Please add]

Results

Please compare the results of produced concrete within existing published research. Line or bar charts can be added. [Please add]

-Can you add some economical analysis of using this material.

Conclusion:

This section must contain implications for research, practice and/or Field: Does the paper identify clearly any implications for research, practice and/or society? Does the paper bridge the gap between theory and practice? How can the research be used in practice (economic and commercial impact), to influence technical policy, in research (contributing to the body of knowledge)? Add something for field professionals. [Please add]

Limitations of the study:

Please add as heading 5 about the limitations of the study.

Author Response

The authors wish to thank the reviewer for his/her comments and suggestions, all of which are addressed in the revised MS. All deletions, additions and changes are shown in green in the new version for readier re-review. The paper has also been thoroughly proofread and a number of typos corrected.

This manuscript is about Sulfate Resistance in Cements Bearing Ornamental Granite Industry Sludge.

The paper is of average quality. Some suggestions, the authors should consider improving the quality of manuscript:

Abstract:

Add something about the results of the research. [Please add quantities] The authors thank the reviewer for this suggestion and have proceeded accordingly.

Introduction:

Please add a list of those materials which have already been used in previous studies for same purpose and now Ornamental Granite Industry Sludge will replace. [Please add in line 86-98] The authors thank the reviewer for this suggestion and have proceeded accordingly.

Against that backdrop, this study provides scientific-technical insight into the effect of replacing 10 % or 20 % cement with granite sludge on sulfate resistance by analysing the mechanical behaviour, porosity and weight of cement pastes made with new blended cements exposed to aggressive environments for different times, along with the microstructural changes taking place (using MIP, XRD, FTIR and SEM/EDX). [Please clarify lines 99-103-Clearly explain what and why you are doing this research] The authors thank the reviewer for this suggestion and have proceeded accordingly.

Experimental

Change heading wth standard formate- ‘Materials and Methods’. The authors thank the reviewer for this suggestion and have proceeded accordingly.

A comprehensive research framework missing- to follow the research is steps are missing. Add framework-flowchart and write this section in stepwise pattern.

Step.1, Step.2 …..[Please add] The reviewer’s suggestions are well taken and have been included in section 2.3, Methodology.

2.1. Materials

Please add details of at least 5-6 materials which were used in previous studies and how authors got an idea to use Ornamental Granite Industry Sludge. Further also explain how it was proved that this material has certain binding properties to replace existing materials upto 10% or 20%. [Please add] This matter is addressed in the Introduction (lines 71-86), which lists some of the most prominent studies on the use of granite sludge as a cement addition. The replacement ratios proposed in this research were based on reports in the literature and chosen to comply with cement type II/A and IV/A requirements.

Results

Please compare the results of produced concrete within existing published research. Line or bar charts can be added. [Please add] This request could not be met because the present findings for cement pastes with new additions are not comparable to those reported by other authors for concrete, in light of the structural and compositional differences involved.

-Can you add some economical analysis of using this material. Unfortunately, that information is not presently available.

Conclusion:

This section must contain implications for research, practice and/or Field: Does the paper identify clearly any implications for research, practice and/or society? Does the paper bridge the gap between theory and practice? How can the research be used in practice (economic and commercial impact), to influence technical policy, in research (contributing to the body of knowledge)? Add something for field professionals. [Please add] The authors thank the reviewer for this suggestion and have proceeded accordingly.

Limitations of the study: The authors thank the reviewer for this suggestion and have proceeded accordingly.

Please add as heading 5 about the limitations of the study. The authors thank the reviewer for this suggestion and have proceeded accordingly.

Reviewer 4 Report

  1. The conclusion of this article lacks the results of sludge used in cement. The author must propose specific application results based on the experimental results.
  2. In Figure 4. That circumstance was directly related to the inverse linear relationship (Figure 4) between flexural strength (FS) and
    pore system properties (total porosity and percentage of macropores). However, the result of selecting 3 test parameters and then using linear regression analysis. Because there are only 3 test parameters. Is it appropriate to choose linear regression analysis? Whether to adopt histogram analysis?

Author Response

The authors wish to thank the reviewer for his/her comments and suggestions, all of which are addressed in the revised MS. All deletions, additions and changes are shown in brown in the new version for readier re-review. The paper has also been thoroughly proofread and a number of typos corrected.

1. The conclusion of this article lacks the results of sludge used in cement. The author must propose specific application results based on the experimental results. The comment has been considered.

2. In Figure 4. That circumstance was directly related to the inverse linear relationship (Figure 4) between flexural strength (FS) and pore system properties (total porosity and percentage of macropores). However, the result of selecting 3 test parameters and then using linear regression analysis. Because there are only 3 test parameters. Is it appropriate to choose linear regression analysis? Whether to adopt histogram analysis? Figure 4 shows the linear correlation between flexural strength/total porosity and flexural strength/macropore volume for the cements studied (OPC, OPC+10GS and OPC+20GS) at two soaking times (56 d and 180 d) in two media (water and sulfate). Medina et al. [1], Aligizaki [2] and Kumar and Monteiro [3] observed the same correlation between those parameters.

[1]   J. M. Medina, I. F. Sáez del Bosque, M. Frías, M. I. Sánchez de Rojas, C. Medina, Design and properties of eco-friendly binary mortars containing ash from biomass-fuelled power plants, Cement and Concrete Composites 104 (2019) 103372.

[2]   K. K. Aligizaki, Pore structure of cement-based materials. Testing, interpretation and requirements, Frist Edition ed., Taylor & Francis, New York, 2006, pp. 388.

[3]   P. Kumar Metha, P. J. M. Monteiro, CONCRETE: Microstructure, Properties and Materials, Third Edition ed., McGraw-Hill, United States of America, 2006, pp. 659.

Reviewer 5 Report

  1. Moderate gramatical corrections required. The corrections are mentioned in the pop-ups when cursor is moved over blue color strike-through & smaller blue triangles.
  2. I have query for the authors, you have made prisms from pastes of OPC, OPC +10% GS and OPC + 20% GS at water/cement ratio = 0.5. Why this w/c ratio? Why not w/c=0.4, which is recommended for making pastes and mortars?
  3. Topic 3, Results should be changed to Results & Discussion.
  4. In Table.2, no units have been mentioned for Flexural strengths of different speciments. Have you evaluated flexural strengths in MPa (mega-pascals)?

Author Response

The authors wish to thank the reviewer for his/her comments and suggestions, all of which are addressed in the revised MS. All deletions, additions and changes are shown in violet in the new version for readier re-review. The paper has also been thoroughly proofread and a number of typos corrected.

  1. Moderate gramatical corrections required. The corrections are mentioned in the pop-ups when cursor is moved over blue color strike-through & smaller blue triangles. The corrections suggested have been included in the revised version of the paper.
  2. I have query for the authors, you have made prisms from pastes of OPC, OPC +10% GS and OPC + 20% GS at water/cement ratio = 0.5. Why this w/c ratio? Why not w/c=0.4, which is recommended for making pastes and mortars? A w/c ratio was chosen further to European standard EN 197-1 recommendations for standardised mortars and because it ensured similar workability of the pastes prepared.
  3. Topic 3, Results should be changed to Results & Discussion. The authors thank the reviewer for this suggestion and have proceeded accordingly.

Round 2

Reviewer 3 Report

This manuscript can be accepted in current form.